# A Model Framework for the Estimation of Animal ‘Suffering’: Its Use in Predicting and Retrospectively Assessing the Impact of Experiments on Animals

**DOI:** 10.3390/ani13050800

**Published:** 2023-02-22

**Authors:** David B. Morton

**Affiliations:** School Bioscience, University of Birmingham, Birmingham B15 2TT, UK; dbmgm2@gmail.com

**Keywords:** assessment, adverse effects, mental pain, distress, severity limits

## Abstract

**Simple Summary:**

Assessing the impact of scientific procedures on animals is not always easy, especially when considering the variety of species, ages and experimental conditions to which research animals are exposed. It is important to do so, as legislation demands that humane endpoints or intervention points be implemented by scientists when appropriate during an experiment. Herein, I describe a scheme that is objective and avoids subjective assessments. It can be applied to many types of experiments and for most species of animals used in research. It has the additional advantage that the type of adverse effect does not have to be specified, e.g., pain, distress, suffering or lasting harm, as it measures the impact of the experiment on the animals, which is more likely to reflect their emotional state. The scheme can also be used to assess the effectiveness of any alleviative therapy.

**Abstract:**

This paper presents and illustrates, with a working example, a hypothesis for the assessment of ongoing severity before and during an experiment that will enable humane endpoints and intervention points to be applied accurately and reproducibly, as well as helping to implement any national legal severity limits in subacute and chronic animal experiments, e.g., as specified by the competent authority. The underlying assumption of the model framework is that the degree of deviation from normality of specified measurable biological criteria will reflect the level of pain, suffering, distress and lasting harm incurred by or during an experiment. The choice of criteria will normally reflect the impact on an animal and have to be chosen by scientists and those caring for the animals. They will usually include measurements of good health such as temperature, body weight, body condition and behaviour, which vary according to the species, husbandry and experimental protocols and, in some species, unusual parameters such as time of the year (e.g., migrating birds). In animal research legislation, endpoints or severity limits may be specified so that individual animals do not suffer unnecessarily or endure severe pain and distress that is long-lasting (Directive 2010/63/EU, Art.15.2). In addition, the overall severity is estimated and classified as part of the harm: benefit licence assessment. I present a mathematical model to analyse the measurement data to determine the degree of harm (or severity) incurred. The results can be used to initiate alleviative treatment if required or if permitted during the course of an experiment. In addition, any animal determined to have breached the severity classification of a procedure can be humanely killed, treated or withdrawn from the experiment. The system incorporates the flexibility to be used in most animal research work by being tailored to the research, the procedures carried out and the species under investigation. The criteria used to score severity can also be used as additional scientific outcome criteria and for an analysis of the scientific integrity of the project.

## 1. Introduction

Morton and Griffiths [1] pointed out that the very first step in any assessment for avoidance or alleviation of pain and distress in animals is to recognise when they are suffering or experiencing adverse states in some way. (I am using the term suffering in a broad way to encompass all adverse physical, physiological and mental states in animals, which may be exacerbated further through some form of cognitive reflection and memory (‘mentation’ as termed by De Grazia [2]). Other ways of looking at suffering include the notion of an impaired wellbeing and measuring the impact of the procedures on an animal. While pain and mental distress may be the commonest adverse states for most experimental animals, there are other ways in which animals experience adverse states that can lead to poor physical, physiological and mental welfare and, consequentially, a poor quality of life. In animal research, the impact of an experiment on an animal’s homeostasis (as measured by deviation from ‘normality’ that is not transient) will most likely reflect the harmful consequences of experimental procedures, as well as the extent to which the animal is suffering [1]. As we humans cannot verbally communicate with animals, this approach is as good as (and probably better than) most other scientific measures. It should be appreciated that all scientific measures of welfare, whether physiological, biochemical, hormonal, behavioural or genomic, need to be interpreted for assessment of the degree of harm that an animal is experiencing, which can be further categorised as a mental or physical state (and the possibility and need to provide some form of alleviation). 

An assessment may be carried out for the purposes of avoidance of suffering (through refinement of the experimental design), to determine the effectiveness of alleviative therapies (e.g., analgesics for pain or anxiolytics for fear) or, in the case of animal research, for assessment of the level of severity (adverse effects) for ethical and legal reasons. Severity comprises two components: the intensity of the adverse state such as pain or distress and, secondly, its duration. Another factor to consider when calculating the overall animal welfare impact of an experiment is any cumulative suffering, for example, inadequate husbandry, in addition to the experimental procedures, the number of times an animal is subjected to a repeated technique (e.g., injections) or procedure or any additional procedures in more complex protocols. Short-term intense stressors may also have long-term impacts, e.g., on the immune system. It is possible to determine the overall animal ‘cost’ in terms of harms in a series of experiments by including the number of animals affected.

In competent humans, the assessment of suffering usually relies on verbal reporting of some sort, whereas in babies and other non-verbal or non-communicating humans and non-human animals, other methods have to be used. In animals, an estimate of the intensity of suffering can be made by measuring the impact of an experimental protocol on an animal, i.e., the degree to which it has deviated from normality compared with a control naïve group kept in a similar or the same way. The duration of any suffering is the second most important component of severity and is far easier to measure objectively, although it is often not specified. The combination of intensity and duration is used to estimate ‘severity’. The advantage of measuring impact by deviation from normal in this way is that it does not require the type of adverse state experienced to be explicit, that is, it does not need to be defined as pain, physical dystress (a state of stressor overstimulation of the endocrine system as first described by Moberg 1985) [3] or mental distress, such as fear, anxiety, boredom, frustration, grief, etc. If there is no measurable impact, then it is unlikely that the animal will be suffering to any serious extent. That said, short-term suffering is very difficult to assess, as many of the usual markers such as weight loss or appetite will not be measurable, and other, usually behavioural, measures need to be used. Consequently, the approach outlined in this paper is not applicable to acute/hyperacute suffering and is more applicable when the duration extends over a day or more. Scorable clinical signs have to be selected in accordance with the cause and the type of suffering involved, which are partly determined by the experimental procedures being carried out and the species. These clinical signs (sometimes mistakenly referred to as symptoms, which are feelings reported by patients to doctors and not observable) in animals must be observable and assessed as to whether they are within the range of normality or whether they are ‘abnormal’. Recognition can be grouped into five categories: relating to appearance, posture, natural behaviour, provoked behaviour and clinical measurements [1]. 

Considerable advances have been made in the details of these sorts of abnormalities and their observational robustness in many areas of research. For example, in one year in one laboratory animal journal, four papers were published in the fields of rat models of osteoarthritis [4,5,6,7]. Three or four decades ago, no mention would have been made of the adverse effects on an animal in an experiment, let alone any measurement of them [8,9,10,11]. This serves to emphasise that with the increasing awareness that pain and distress in animals is important, the measurements used are become increasingly more accurate, reliable and generally accepted. Moreover, the Federation of European Laboratory Animal Science Associations (FELASA) provides comprehensive guidelines on the classification and reporting of severity experienced by animals used in scientific procedures [12].

### What to Measure?

In a research setting, it has to be decided what measurements to take to assess any suffering. It is always better to use several criteria to ensure a better estimate, as just one or two may be misleading. It may be possible to carry out measurements of the body’s reaction to adverse effects such as effects on the sympathetic nervous system (catecholamine levels and their corresponding end-organ responses, e.g., heart rate, blood pressure and acute-phase proteins) or the pituitary–adrenal cortex axis (particularly adrenal corticosteroids). However, these tests can be expensive and may themselves require invasive blood sampling methods, whereas the behaviour of an animal can be simply observed without cost, including how an animal responds to a particular stimulus (provoked behaviour). Furthermore, some of the affected parameters may be compounded by an inherent variation and vary according to the time of day, the time at which an animal is observed or sampled and the specific experimental procedures being carried out. The time and method of sampling, as well as behavioural observation, have to be practical. For example, in terms of time, one would expect a greater change in pain and distress parameters around the time of surgery rather than 2–4 weeks later. For experimental arthritis, adverse feelings of pain such as dull aches may persist for weeks and not just hours or days as for a surgical procedure; therefore, measuring body weight may be more appropriate. For an experiment with several techniques or procedures being carried out over time, several time points within the whole experimental period may have to be scored to develop an accurate picture of the overall ‘suffering’ from the animal’s viewpoint. In conclusion, the time of observation and the key criteria to be measured have to be carefully selected. 

In EU Directive 2010/63/EU [13], harm or the adverse effects are defined as “any pain, suffering, distress or lasting harm equivalent to or higher than that caused by the introduction of a needle in good veterinary practice” experienced by an animal as a result of the scientific procedures performed ([13] Art. 3.1). There is an obligation for an assessment of the harm likely to occur as a result of the experimental protocols to be conducted at the outset of a research project and for this to be recorded on the project licence as the severity classification for a procedure or series of procedures, i.e., for the whole project. Severity may also have to be recorded in hindsight as the harm that actually occurred, referred to in the Directive as a “Retrospective assessment” ([13] Art. 39); in specific cases, this is a legal requirement. If an animal is to be reused, then the severity of any previous use has to be determined ([13] Art. 16.1. (a)). Finally, almost all international legislative measures to control animal experimentation worldwide require the successful application of the Three Rs. This paper deals with the aspect of refinement and provides a model whereby animal suffering can be recognised, reduced, reviewed and even terminated on objective grounds when necessary.

This model (first drawn up in the UK prior to the revision of the Directive [13] in 2010 and presented at various seminars) concerns measuring severity retrospectively and comparing it with the predicted severity in order to determine if the severity classification is being or has been exceeded. It is based on the author’s practical experience as a laboratory animal veterinarian and close observation of experimental animals in a variety of research settings. In addition, FELASA has produced some valuable and complementary guidelines (12) that expand further on this theme and provides several more comprehensive and complicated examples. In this paper, I have tried to provide a more practical and accessible approach that is simpler and easier to apply through the selection of the key criteria that fit the purpose of assessing the level of harm and that can be applied to all vertebrates and even invertebrates.

## 2. Methodology and Results

The first stage is to identify the criteria that can be used to assess any harm and then to select from those the ones that commonly occur and are easy to score. The second stage is to select the time at which to score those criteria. There is no limit to the number of criteria scored or the number of time points chosen, but in order to reduce workload, only the key and most relevant criteria should be used. These key criteria should also best reflect an animal’s adverse state and not simply repeat other closely correlated signs. In addition, the measured signs should reliably indicate severity, be robust, subject to little inter- or intraobserver variation, and be economical and easy to measure. Furthermore, the criteria measured should be independent of each other, for example, a failure to eat and body weight are usually closely interdependent.

The aim of this semiquantitative assessment is to be able to compare predicted and actual suffering and to express these estimates mathematically. In the following example, the deviation from normality or intensity is determined for each independent criterion for each animal in the group at a set time. This figure is then averaged to obtain a score for that criterion. The average score per animal is made up of the averages for each criterion. It is then possible to compare, for example, different experimental groups at the same time point, as well as at different time points in an experiment. The easiest way to understand this is to provide a simplified working example.


**Example: A Hypothetical Comparison of Two Dose Groups in a Toxicity Test in Rats**


In a study of drug safety and effectiveness, several doses were given, and the rats were observed for adverse effects. Pilot studies and previous data with respect to closely related compounds had shown the chemical’s likely adverse effects, and the relevant criteria were chosen according to the criteria described above. In this example, only two dose groups and two criteria were compared at the same time point. The project licence for the work had predicted the severity classification, and severity limits had been set for individual animals in a dose-level group. For the Low-dose group, the severity limit was mild, and for the High-dose group, the severity limit was severe. The accuracy of these severity limit predictions can be determined by assessing the actual severity outcomes in real time as illustrated below.

### 2.1. Choice of Criteria

Affected animals were predicted to show signs, to varying degrees, of lethargy, fever, reduced appetite, a failure to groom or preen, lachrymal secretions from the eyes and nose, diarrhoea and occasional vomiting. The control group (in this case, drug vehicle only) kept under similar conditions was not expected to show any adverse effects and would be classified as the ‘normal comparator’ for the dosed groups. Among these clinical signs, body temperature (Criterion 1) and body weight (Criterion 2) were the easiest and most accurate to measure.

### 2.2. Scoring the Criteria

The next step was to classify the criteria to reflect the ‘intensity’ of the adverse state, i.e., the impact on the animal based on the extent to which the two criteria deviated from normality (see Table 1 and Table 2). Three major bands were chosen to reflect the three recognised legal severity categories of mild, moderate and severe. However, in practice, there are two other recognisable and necessary categories: no change (normal variation) and excessive change that is beyond the upper severity limit (i.e., more than severe and long lasting, which has to be described and be quantitatively and qualitatively defined and would include death). These bands were also scored, resulting in a total of five bands. For body temperature, the five bands corresponded to no significant change (0–0.2C, i.e., normal), an increase of 0.2 to 1C (mild), and increase of 1 to 2C (moderate), an increase of 2 to 3C (severe) and an increase of more than 3C above normal (greater than severe). 

For body weight, the five bands are no significant change (0–5%, i.e., normal); a mild change of 5–10%, a moderate change of 10–20%, a severe change of 20–30% and a loss of more than 30% (greater than severe). Each of the five recognisable bands for each criterion is assigned a score of 0, 1, 2, 3 or 4, and in this way, the scores are converted into a mathematical model for analysis. Each animal is scored at the same time point for each criterion at the same time of day in an attempt to keep everything constant except for the measurement. This figure is then averaged to obtain a group average for that criterion at that specific time. 

Part A in Table 1 (body temperature, low dose) shows that two animals scored within the range considered normal, and three animals had a rise of between 0.2 and 1C. This resulted in a group score of 3 and a group average of 0.6 (i.e., total group score (3) divided by 5 animals) for that criterion. 

For body weight (body weight (low dose), Part B in Table 1) four animals lost between 5 and 10% body weight, and one animal lost between 10 and 20% body weight, resulting in a group score of 6 (0 + 4 + 2 + 0 + 0) and a group average of 1.2 (i.e., total group score (6) divided by 5 animals). The same process is now applied to the high-dose group for the two criteria (see Table 2).

The scores for the calculation of actual overall impact assessment can now be performed by summing the two criteria to derive an average impact intensity score for each dose level.

### 2.3. Low-Dose Group


$$\begin{array}{ll} Overall\;score & :=\;\frac{Number\;of\;animals\;\times \;\left(group\;average\;criterion\;1\;+\;group\;average\;criterion\;2\right)}{Number\;criteria\;used}\\ & \;\;=\frac{5\;\times \;\left(0.6\;+\;1.2\right)}{2}=\frac{5\;\times \;1.8}{2}=4.5 \end{array}$$


Therefore, the average impact score for each criterion for each animal in the low-dose group is 4.5/5 = **0.9**.

### 2.4. High-Dose Group


$$\begin{array}{ll} Overall\;score & :=\;\frac{Number\;of\;animals\;\times \;\left(group\;average\;criterion\;1\;+\;group\;average\;criterion\;2\right)}{Number\;criteria\;used}\\ & \;\;=\frac{5\;\times \;\left(3\;+\;3.2\right)}{2}=\frac{5\;\times \;6.2}{2}=5\;\times \;3.1=15.5 \end{array}$$


Therefore, the average impact score for each criterion of each animal in the high-dose group is 15.5/5 = **3.1**.

## 3. Interpretation of the Results

The data obtained from this example can be used to determine whether the severity limit has been exceeded during the study, as well as whether the overall severity classification for the project has also been exceeded.

### 3.1. Severity Limit

The severity limit is imposed to protect the individual animal; any animal that exceeds the severity limit on the average of the specific criteria should be reported to the project licence holder to take some sort of action. For example, the animal can be withdrawn from the study or, in some circumstances, attempts can be made to alleviate the suffering. An alternative is to contact the competent authority to change the limit and for the study to continue as planned.

### 3.2. Severity Classification of the Project

A comparison can now be made with the prediction of mild severity in the low-dose group and severe in the high-dose group. In theory, the average score for each animal in a mild band should be 1; however, the actual average was 0.9. Therefore, in the final analysis for that time point, the suffering was less than predicted. Using the same approach, the average score for each animal in the high-dose group should have been 3 (severe) but it was actually 3.1. Therefore, in the high-dose case, the severity classification was exceeded. As the severity limit applies to an individual animal (see [13]. Dir.Annex VIII), some of these animals should have been killed or withdrawn from the study, although their scores should still be recorded.

## 4. More Complex Experiments: Continued Use and Reuse (Directive 2010/63/EU, Art 16.1 (a))

If a project is more complex with several phases comprising a continued use, each phase can be scored separately and assigned a severity limit, e.g., surgery to implant a monitor followed by a treatment and removal of the monitor. The cumulative severity would then be the severity of each phase of the project, i.e., the severity classification for the whole project. This may be useful scientifically, as a grossly abnormal animal during one phase may not yield reliable data at a later stage because the results could have been confounded by physiological, homeostatic or behavioural responses during an earlier phase.

In terms of reuse, no animal may be reused if it has experienced suffering of mild or moderate severity during a previous use; this scheme will be useful for such assessment. 

## 5. Discussion

### 5.1. Legal Aspects

Current EU [13] and UK [14] legislation requires project licence holders to conduct an ongoing assessment of the levels of pain, suffering, distress and lasting harm (severity) experienced by animals in an experiment. It also requires that any such suffering be reduced to the minimum necessary to achieve the scientific objective. The scheme described in this paper will help to do this for subacute and chronic experiments (i.e., more than 2 days) but not always for acute experiments, in which the intensity of suffering may exceed the severity limit but not result in any significant physiological or physical changes that can easily be observed and scored. These hyperacute changes, indicating pain and distress, require a different approach such as observing the behaviour of the animals (e.g., vocalisation, escape behaviours, focal attention to a particular body site, etc). It would also be possible to measure blood hormone hormones (e.g., a rise in corticosteroid and/or catecholamine levels) and end-organ responses (e.g., increases in heart rate, blood pressure and respiration); however, these are complicated by the measurement procedures involved, e.g., handling and blood sampling, which in and of themselves, can cause extra suffering [15].

### 5.2. Choice of Signs to Score

The framework outlined above allows the licence holder to use whatever signs are most appropriate for the specific experimental procedures being carried out. They can be clinical signs, a change in normal behaviour, exhibition of abnormal behaviour or an experimental variable that itself may indicate an adverse effect. The key points are that the signs being scored should be robust in the sense that they reliably indicate a scorable change; any change is easy and convenient to observe; scoring does not cause any further suffering to the animals concerned, i.e., measurement should be non-invasive; and is economical to implement. It is best if the signs also reflect some biological significance; for example, a ruffled/harsh/starey coat is a more general sign of poor wellbeing than measuring tumour size or joint pain, which are specific to the experiment. General signs may be of help in diagnosing an unspecified adverse state, whereas specific signs can be used to make a better scientific severity assessment and may also be used to help with the scientific objectives.

Some signs can relate more to mental distress than pain, and these two states, i.e., pain and distress, often go hand in hand; all painful states are likely to cause mental distress, but not all signs of mental distress indicate pain.

### 5.3. Number of Scorable Signs

How many scorable signs are needed to make an assessment? In one sense, the fewer signs to score, the easier; therefore, it is useful to try to restrict the number to those that are most important or are most likely to reflect animal suffering. The signs to be scored can be evaluated using a statistical elimination/reduction analysis and the minimum number of signs that closely correlate with the final severity outcome. This may be a circular argument and open to bias, but the overall direction of changes and their magnitude make it less likely to be wildly inaccurate. Using the approach described above, any number of signs can be scored cumulatively and scored in pairs or trios, e.g., A + B, or A + B + C, etc., to provide the best fit; that is, A should normally always accompany B, B should not normally be measured in combination with C, etc.

The final crucial assessment is to compare and rank the experimental group(s) with the control group. The control group may be a scientific control (e.g., vehicle only or sham-operated); the most valid type of control would probably be to compare the experimental group with a group of naïve animals kept under the same conditions. Scoring the ‘extra’ severity or suffering is a true indication of the impact of the experimental procedures on an animal. It is worth noting that even the scientific control may sometimes experience some pain and distress, e.g., injection of vehicle only or sham surgery. It is therefore important to always measure and score the control group as well as the experimental group. A comparison with naïve animals kept under the same housing and husbandry conditions provides the most objective indicator of all extra adverse effects experienced by the animals in the experiment, although it will not include any mental distress that may be caused by the housing and husbandry alone.

### 5.4. Categories of Adverse States

Adverse states inevitably include both a physical and a mental component; for example, all painful states will also have a mental component of suffering. Distress, on the other hand, is more difficult to determine and quantify and can be broken down into two types. The classical type described by Moberg [3,16], where there is a marked perturbance in the pituitary–hypothalamic axis that affects end-organ responses such as the adrenal cortex, thyroid and reproductive organs can lead to a marked loss of homeostasis and, ultimately, a failure to thrive. This type of stress response is termed *physiological dystress*. The other sort of distress, which was originally described by Selye, involves behavioural changes and is more mental than physical. It can be caused by negative emotional states of, e.g., frustration, boredom, fear or anxiety, and is usually transient, although not always; this type of distress is termed *mental distress*. The choice of controls mentioned in the preceding paragraph dealing with naïve animals will inevitably include a minimum of mental distress as a result of the housing and husbandry, e.g., frustration and boredom. Extra suffering as a result of the procedures is the difference between the control group and the experimental group(s), e.g., pain and fear. This further illustrates the difference between any physiological dystress and any mental distress.

### 5.5. Change in Score Direction

Some changes in scorable signs may be positive or negative and can go in either direction depending on the experiment and the species being scored. For example, in experimental studies on infection, an increase in body temperature is normally seen in most species, but for those species with a high metabolic rate, e.g., small rodents such as mice, infection may lead to a decrease in temperature; exactly the same principle applies to scoring the severity deviation from normality [1,17,18]. In terminally ill animals, the body temperature will gradually fall if they are left to die. When using body weight as a measurement, the age and maturity of an animal has to be taken into account. That is why the control should be chosen carefully. A failure to eat will lead to body weight loss depending on the level of intake, but in growing animals, the comparison should be age-matched, as a slowing of growth may be just as important. Using body weight to measure tumour growth may result in appetite being decreased or increased, whereas body weight may increase or decrease or not change. It has to be coupled with other signs such as tumour size. Body weight may decrease as a result of lowered intake or increased metabolic rate, or an increase in body weight may be observed as a result of fluid accumulation, e.g., ascites with liver tumours or decreased activity. All confounding factors need to be taken into account and interpreted biologically when selecting appropriate clinical signs of suffering.

### 5.6. Scoring of Death

Death should always be scored as a criterion in its own right, as it represents major change (usually unexpected) and should considered a serious adverse effect regardless of deviation from normality. If death is expected, stringent efforts must be made to predict it and to apply a humane endpoint, as death often indicates prior hyperacute suffering of some sort [19]. However, death can also occur very rapidly between inspections. The legislation permits death as an endpoint in exceptional circumstances [20].

### 5.7. What Is Normal?

Sometimes it can be challenging to decide what is the normal range for a particular sign, as there is so much biological variation. Each animal is scored as an individual and not as a group; therefore, investigators look for a change in the individual rather than an absolute number or standardised qualitative estimate. Depending on the species, change relative to normal can vary in rodents from 0–10% for body weight depending on the time of day, whereas for other species, the range is much smaller. This is the reason why scoring should take into account the circadian rhythm of animals and ideally be carried out at the same time each day. An animal’s reaction to its housing and husbandry vary as much as the husbandry, the housing and the personnel. That confounded by individual biological variation means that ‘normal’ will inherently have some variation, which is why +/−5% leeway is given for that category.

In the examples I have chosen for this study, I have given a score for each level of change that correlates with severity classification in the legislation. The classification of normal can be difficult in some species due to the natural variation in some species, especially in non-mammals. Normal can vary with ambient temperature (poikilotherms such as retiles amphibia, fish, including and the mole rat) and season of the year (e.g., body weight in migratory birds); however, for laboratory mammals kept under research conditions and husbandry, ‘normal’ parameters are relatively stable for a given age and sex. The normal baseline forms the basis for assessing the impact of an experiment on an animal. The degrees of change above or below the normal threshold are set to reflect the severity classification from ‘mild’ to greater than ‘severe’, which includes death; although changes greater than severe are not permitted, they will inadvertently occur between scoring inspections unless an adequate safety margin is implemented (see the section above on scoring death).

### 5.8. Assessment of the Severity Limit and Severity Band

The average score for an individual animal in a group at a particular time is the sum of all the scores for a specific sign divided by the number of animals. This score is then added to the scores for all the other signs, which provides an estimate of the actual severity being experienced by that animal at that time. For the total score in an experiment, all the scores for all the animals in that group are added together, and that sum can then be used to obtain the average score for each animal in that experimental group. This figure can them be used to determine any breach of the project severity classification, which can be an overestimate or an underestimate. However, for the application of the severity
limit and the implementation of a humane endpoint, it is the estimated suffering of an
individual animal that is important. *The project severity classification*, on the other hand, is based on the overall average for the worst-case scenario of the project. The estimation of the level of suffering (severity) can then be used to corroborate the estimated projected severity of the project.

### 5.9. Inspection Intervals

It is a legal requirement that animals be inspected for health every day ([12] Art 13.1. (c)). Animals used in an experiment should be checked more regularly depending on to how likely they are to undergo a change in their severity classification; the frequency of inspection should be increased for higher severity levels to three to four times or more daily or if there is a likelihood of exceeding the upper severe category during the night and outside normal working hours (e.g., weekends) (see unpredicted endpoints in Ashall and Millar, 2014) [21].

### 5.10. Uses of Scoring Results Other than Severity Estimates

Ongoing contemporaneous scoring of the signs can help to achieve the goals of ascertaining when an animal has exceeded an allocated individual severity classification or limit, in addition to helping to define a humane endpoint, scientific endpoint or intervention point depending on which is being sought. It will also help in any retrospective analysis of any suffering experienced by animals, as well as the effectiveness of any alleviating strategies. Scoring the adverse effects of scientific procedures during an experiment will help in the validation of the scientific data being measured, as confounding covariables may negate or modify the interpretation of the scientific results. It must always be remembered that an animal’s physiological responses to the experimental procedures being carried out are likely to affect the scientific data being harvested and may confound the interpretation of the data. Furthermore, these physiological responses may even be of greater significance than any alleviative treatments; therefore, attempts should always be made to treat pain and distress.

*Training:* For some clinical signs used for scoring, e.g., behaviours, appearance, stance and posture and provoked responses, it is often vitally important that there is consistency between those scoring the signs; this may require some serious training. Such training should be extended to all appropriate staff, including those staff working at weekends, holidays and out-of-hours during the evening and the night.

Finally, it should not be forgotten that such an approach may be useful to assess positive mental states, as well as negative ones, although the choice of specific measures will obviously vary, e.g., use of any enrichment or attractions and time spent playing in young animals.

## 6. Conclusions

This paper describes the principles of a simplified system of analysis of the impact of an experiment on an animal that is objective, robust and reproducible. It will permit a contemporaneous assessment of the adverse effects and a scheme for the implementation of the severity classification limit. It will also provide a mechanism for the retrospective assessment and accuracy assessment of the predicted severity classification.

## Figures and Tables

**Table 1 animals-13-00800-t001:** (A) The proposed scoring matrix for body temperature (low dose). (B) The proposed scoring matrix for body weight (low dose).

**A**					
**Change**	**<0.2**	**>0.2 to 1C**	**>1 to 2C**	**>2 to 3C**	**>3C**
**Control**	**Normal range**	**Mild**	**Moderate**	**Severe**	**>Severe or death**
**SCORE:**	**0**	**1**	**2**	**3**	**4**
1	X				
2		X			
3	X				
4		X			
5		X			
Total score	(2 × 0)	(3 × 1) = 3	0	0	0
**B**					
**Change**	**0 to 5%**	**>5 to 10%**	**>10 to 20%**	**>20 to 30%**	**> 30%**
**Control**	**Normal range**	**Mild**	**Moderate**	**Severe**	**> Severe or death**
**SCORE:**	**0**	**1**	**2**	**3**	**4**
1		X			
2		X			
3		X			
4			X		
5		X			
Total score	0	(4 × 1) = 4	2	0	0

Group total = 0 + 3 + 0 + 0 + 0 = 3; group average = 3/5 = 0.6.

**Table 2 animals-13-00800-t002:** (A) The proposed scoring matrix for body temperature (high dose). (B) The proposed scoring matrix for body weight (high dose).

**A**					
**Change**	**<0.2**	**>0.2 to 1C**	**>1 to 2C**	**>2 to 3C**	**>3C**
**Control**	**Normal range**	**Mild**	**Moderate**	**Severe**	**>Severe or death**
**SCORE:**	**0**	**1**	**2**	**3**	**4**
1				X	
2				X	
3					X
4			X		
5				X	
Total score	0	0	1 × 2 = 2	3 × 3 = 9	1 × 4 = 4
**B**					
**Change**	**0 to 5%**	**>5 to 10%**	**>10 to 20%**	**>20 to 30%**	**>30%**
**Control**	**Normal range**	**Mild**	**Moderate**	**Severe**	**>Severe or death**
**SCORE:**	**0**	**1**	**2**	**3**	**4**
1				X	
2					X
3					X
4			X		
5				X	
Total score	0	0	1 × 2 = 2	2 × 3 = 6	2 × 4 = 8

Group total = 0 + 0 + 2 + 9 + 4 = 15; group average = 15/5 = 3.0.

## Data Availability

Not applicable as a theoretical concept.

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
