# Peer review of "A Model Framework for the Estimation of Animal ‘Suffering’: Its Use in Predicting and Retrospectively Assessing the Impact of Experiments on Animals"

_animals, 2023, doi:10.3390/ani13050800_

Round 1

Reviewer 1 Report

In this manuscript, a model framework for estimating animal suffering is proposed, based on a ‘mathematical’ estimation that accounts for both the impact on animals, and the duration of said impact. There are several caveats with this sort of approach, of which, however, the author is well aware.

My first impression is that it is not clearly explained when and under what circumstances to apply the proposed approach. If it is for cage-side daily assessment, I believe it is overly complicated, like clinical score scoring with extra steps, and quite hard and complex to apply in a satisfactory way. However, I can see value in it if applied to follow closely and in detail the extent to which a new animal model of unknown phenotype and severity impacts animals, and therefore inform subsequent harm-benefit appraisals, as well as planning of refinement measures for said model. I suggest this application is duly explored.

There are a few concepts that I had trouble following, despite working on laboratory animal welfare, which means that the average reader might struggle even more. One in particular is the concept of severity “band” and “banding”, which is not, as far as I am aware, standard terminology. Moreover, some of the examples proposed bands setting differences as meaningful when they were within the standard deviation of the proposed parameters (e.g. 0.2 °C variations in body temperature). Also, as the author is well aware, actual severity classification is done at the level of the individual animals, reason why I was quite puzzled with the proposed approach of averaging severity for a given group of animals (e.g. lines 238-244).

I suggest that the author consults the FELASA report by Smith et al 2018 (DOI: 10.1177/0023677217744587), which provides useful information on severity classification. This report covers extensively the process of severity classification, when assessing both prospective and actual severity, for which these questions are central: “What will the animals experience? How much suffering might it cause? What might make it worse? How will suffering be reduced to a minimum?” Answering these questions allows to depict both best-case and worst-case scenarios, and plan and carry out procedures accordingly. Based on this, prospective severity of a procedure is set based on the group of animals that will be worst off, so any assessment should focus primarily on these ones (say, untreated controls in a study on infection).

Aware of the subjective nature of most clinical assessment, the author proposes the use of quantifiable physiological variables. It should be noted, however, how the process of measuring said variables can impact animals and the readout (e.g. see Blenkus et al 2022 DOI:  10.3390/ani12020177 for the impact of handling on body temperature, and how it can be avoided).

In regard to the text itself, I have the following comments:

Ln 10-11 – I do not think it is the case that legislation demands humane endpoints to be implemented in proportion to benefits, but rather that harms, as a whole, are weighed against benefits. HE, as the author is very well aware, are rather the refinement approaches that can prevent excessive, unnecessary harm.

Ln 46-50 – as pointed out in one of the references provided (Moberg 2000) there is adaptation to changes in homeostasis, always at a cost of welfare, which is hard to recover when intensity*duration of stress goes beyond the adaptability of an animal. Is the author referring to transient changes, or the ones that are overtly seen, which tend to manifest only when the animal is beyond the threshold of being able to hide them?

Line 59 – add “of suffering” after “An assessment may be carried out for the purposes of avoidance”

Line 67 – replace “repeat” for “repeated”

Line 83-85 – Please mind that short term but intense stressors can have long-lasting impact on both mental and physical welfare.

Line 156 – animals of which species?

Line 166-167 – this scenario is really severe, and I would say even beyond the established upper threshold, because it would only be observable following unrelieved prolonged severe pain or suffering. Also, chromodacryorrhea is more likely observable in rats, might as well refer to the species in question. Also, personally I would consider seeing overt “red tears” around the eyes in and of itself a criterion for euthanizing animals. Georgia Mason (2004, DOI: 10.1177/026119290403201s25) has, however, proposed a scoring system for milder signs of chromodacryorrhea, based on small spots on the nose, indicative of mild to moderate stress.

Line 229-230 – exceeding the predefined upper threshold of severity would not be an admissible scenario. Is the author referring to a pre-established limit that is below said threshold?

Line 334 – suggest replacing “animal” for “animals”

Line 344-348 – I would prefer that the author stresses even more that spontaneous death is not an admissible outcome, as it is neither ethically acceptable nor scientifically admissible and, while not strictly illegal (the directive just advises that “death as the end-point of a procedure shall be avoided as far as possible”) as lab animal welfare scientists we should push standards to go beyond regulatory compliance.

Line 366-369 – I had a hard time understanding the described process. I would not trust researchers and animal care staff to find the application of this method appealing.

Line 382 – While I use similar terms to categorize types of endpoint, these are not standard terms. Perhaps refer to the proposed categorization of Ashal and Millar (2014, DOi: 10.14573/altex.1307261)

Further comments

Table legends should be sufficiently detailed to allow interpretation of the tables without having to scan the text. They are documents in of themselves.

While a romance language speaker myself, I would advise against using Latin locutions, such as Nota Bene, and suggest replacing them with “It should be minded that”, or something similar.

Reviewer 2 Report

In this article, the author presents an original model for estimating animal suffering in experimental situations. The term suffering is to be taken here in a broad sense, including any form of physical or mental harm, which could also be described as an impairment of well-being.

After a reminder of the definitions of the key concepts, the methodology is presented and then illustrated using an example representative of a common experimental situation. The proposed methodology is easily transposable to other studies, with the notable exception of studies involving a situation of acute suffering, which is not in the scope of the manuscript.

Relevant advice is given to help the reader define the criteria to be observed, preferably independent of each other, and in a sufficiently limited number to avoid extra workload. The number of criteria should be increased in pilot studies and then reduced as results are obtained for subsequent studies.

The main advantage of this methodology is that it provides quantifiable and mathematically manipulable indicators for groups and individuals. It is also rightly pointed out that for the application of a severity limit during a procedure, only the individual data should be considered.

Overall, this evaluation framework is an interesting tool both for the prospective evaluation of the severity of a procedure as required for a project licence, and for the retrospective assessment of this severity.

However, it should be recalled that the FELASA/ECLAM/ESLAV working group has issued useful recommendations for assessing the severity of a procedure that are complementary to those developed in the manuscript (see Smith et al. Lab Animals 2018 DOI: 10.1177/0023677217744587).

The value of a clear and rigorous methodology is also to encourage refinement of procedures. This aspect would need a line or two with respect to the implementation of the 3R required by UK and EU legislations.

A final recommendation would be that, for the purposes of scientific publication but also for public transparency, the scientific data acquired during a study should be systematically accompanied in the article by the severity data collected. The ARRIVE guidelines should be updated in this sense.

Author Response

Thank you for your helpful comments.  I have accepted them all and changed the manuscript (see attachment)

Reviewer 3 Report

The reviewed manuscript describes a mathematical model for the evaluation of suffering of experimental animals.

 Comments:

Page 4, line 178: “excessive change (i.e. more than Severe or Death)”:  What means this? More than severe suffering? More than death? Death without suffering is the lowest category (terminal). Severe suffering leads to humane endpoints (death) and is therefore less burden than long lasting severe suffering.

 Page 5, Table 1: There is no explanation for the graduation. It is known that body temperature in mice can vary between sexes and activity phases around 1 centigrade. Or the body weight. Loss of more than 20% body weight is a reason to set a humane endpoint.

 Page 6, line 230: The end of the sentence is missing.

Additional comments:

1. What is the main question addressed by the research? Evaluation of cummulative animal burden in experiments.

2. Do you consider the topic original or relevant in the field, and if so, why? The topic is not new.

3. What does it add to the subject area compared with other published material? The author suggests a mathematical model to evaluate the burden of animals in an experiment based on an individual scoring system. The author extended the legal categories terminal, minor, medium and severe burden, introduced a subjective scale and a formula.

4. What specific improvements could the authors consider regarding the methodology? In my opinion there is not really an advantage of the suggested system. It is an abstract try to make the burden evaluation fitting for all species and all kind of experiments. Which will not be possible.

5. Are the conclusions consistent with the evidence and arguments presented and do they address the main question posed? Essentially yes. But the question is, who needs such an abstract system of animal burden evaluation.

Author Response

Thank you for your thoughtful comments.  I have responded to them all and elaborated within the manuscript as appropriate.

Round 2

Reviewer 1 Report

The author addressed most of my comments in a meaningful way, and provided sound arguments for those somewhat disputed. As regards those for which I continue to have divergences (namely averaging severity across different animals) not only I do not think such divergences should prevent publication, but also believe that it is sensible to trust readers on making their own mind based on the arguments presented in the text.

I therefore support the publication of this manuscript. 

(PS - please mind the typo in line 111, "expereinced")